# Representation of Landscape and Ecological Vision in Miyazaki's Filmography

Cristiana Bartolomei [1,*], Alfonso Ippolito [2] and Davide Mezzino [3]

1 Department of Architecture, Alma Mater Studiorum University of Bologna, 40136 Bologna, Italy
2 Department of History, Representation and Restoration of Architecture, Sapienza University of Roma, 00185 Roma, Italy; alfonso.ippolito@uniroma1.it
3 Faculty of Cultural Heritage, International Telematic University Uninettuno, 00186 Rome, Italy; davide.mezzino@uninettunouniversity.net
* Correspondence: cristiana.bartolomei@unibo.it

**Abstract:** This study analyzes the central role of landscape in Hayao Miyazaki's films. The depiction of landscape in Miyazaki's films goes beyond mere visual backdrops in order to convey deep symbolic meanings and to foster an empathic connection between the viewer and the world depicted. The renowned Japanese animator, filmmaker, screenwriter, draftsman, manga artist and film director has strongly promoted environmental awareness in his productions by paying close attention to the depiction and visualization of landscape dynamics, using details, and real and invented elements to create an engaging visual experience. The landscapes also take on emotional, metaphorical dimensions, reflecting the emotions and inner thoughts of the characters. Through an in-depth critical analysis of eleven selected films, the proposed research identifies the character-defining elements adopted by Miyazaki to stimulate reflection on a sustainable combination between urban development and the preservation of natural elements, as well as increasing focus on the beauty of the landscape, thereby highlighting the importance of its preservation. The relevance of this research is to understand Miyazaki's approach to creating representations of natural elements and how he has managed to combine them with the plots of his various films, indirectly stimulating environmental awareness and fascination with nature in its different forms.

**Keywords:** landscape; video animation; representation; Miyazaki; sustainability; ecological vision

## 1. Introduction

The European Landscape Convention defines landscape as "an area, perceived by man, whose appearance and character are the result of the combined action of natural and/or human factors" [1]. This definition reflects the idea that landscapes evolve over time as a result of the interactions between natural and human forces.

Hayao Miyazaki, the renowned Japanese director, scriptwriter and draughtsman of all his works, fits this definition perfectly, as his films explore environmental themes such as the relationship between man and nature, love of nature, pollution and the depletion of the natural world [2]. The contexts and plots of his films always have an educational aspect closely linked to ecology. For Miyazaki, man is at the centre of the narrative, but always at his side are nature and man-made spaces, both in harmony (representing good) and in opposition (representing evil), until a balance of mutual respect is achieved between the individual and the genius loci.

Ecology has been a major theme in the films produced by Studio Ghibli (the production company founded in 1985 by Hayao Miyazaki together with Toshio Suzuki, Isao Takahata and Yasuyoshi Tokuma) for decades, as evidenced by the name of the studio itself, which derives from the Saharan sirocco wind.

For instance, Miyazaki takes time to reflect on the culture of the "evergreen forest", which is gradually disappearing: "When man upsets the balance of the world, the forest

makes great sacrifices to restore it". This quote from Miyazaki's film "Nausicaä of the Valley of the Wind" is one of many that inspire contemplation [3]. His films serve as an ode to nature and provide insight into mankind's role in relation to the natural world. Natural elements and landscapes have always played a central role in Miyazaki's storytelling, allowing him to raise awareness of environmental issues.

In his films, Miyazaki has demonstrated an incredible ability to create imaginary worlds that capture the imagination of viewers of all ages. His works such as "Princess Mononoke", "My Neighbor Totoro", "Howl's Moving Castle" and "Spirited Away", among others, have become icons of cinema and have influenced generations of viewers worldwide [2]. For instance, the 1988 film "My Neighbor Totoro" played a fundamental role in highlighting the importance of the *satoyama*—the traditional Japanese rural landscape of the 1950s—and supporting its preservation. The *satoyama* is depicted with rural settlements, gardens, rice and tea plantations, and with forests in the background (Figure 1).

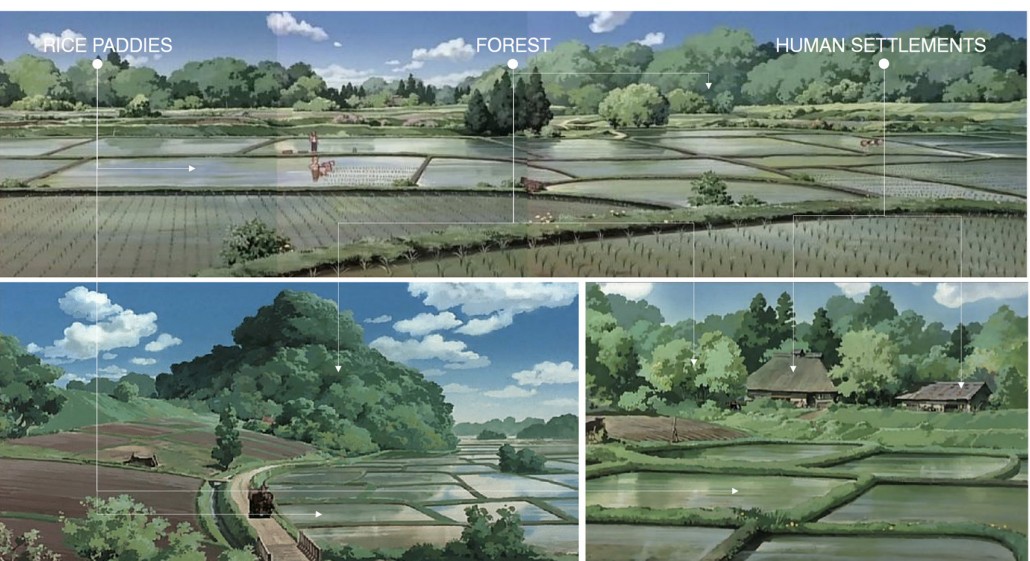

**Figure 1.** Screenshots from the film "My Neighbor Totoro", 1988, showing characteristics of satoyama landscapes; rice paddies, forests and human settlements. Images processed by authors.

Miyazaki's depictions of landscapes evoke and illustrate the importance of living in harmony with nature, even when overwhelmed by technology. This topic is, for example, addressed in the film "Castle in the Sky" (1986), which envisions a super-technological city that peacefully coexists around a massive tree, an ideal combination of science and nature (Figure 2).

Additionally, in-depth attention is dedicated to representing and visualizing landscape dynamics. Each film is marked by a focus on landscape details, creating rich and immersive settings that act as a character in and of themselves. The landscapes painted by Miyazaki go well beyond mere visual backgrounds, conveying deep symbolic meanings and fostering an empathic connection between viewers and the depicted world. Landscapes often play an active role in determining the plot's development and the characters' evolutions [4]. For example, in "Princess Mononoke", the struggle between the forest gods and human industry is reflected in the landscapes where animals, plants and spirits reside. The contrast between pristine forests and the wounds inflicted by humans creates a central conflict in the story, raising questions of sustainability and coexistence between humans and nature (Figure 3). The representation of landscape in Miyazaki's films also stands out for the attention given to the temporal dimension and the natural cycle of the seasons. Landscapes evolve and transform over time, reflecting the cyclical nature of life and nature itself [5].

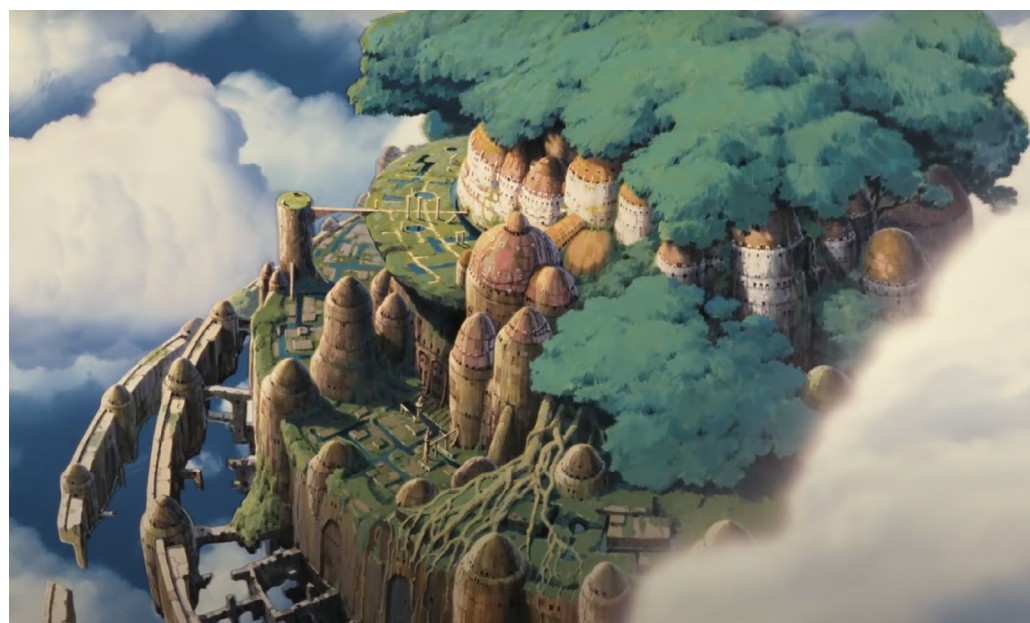

**Figure 2.** Screenshot from the "Castle in the Sky" film, depicting Laputa city. Image processed by authors.

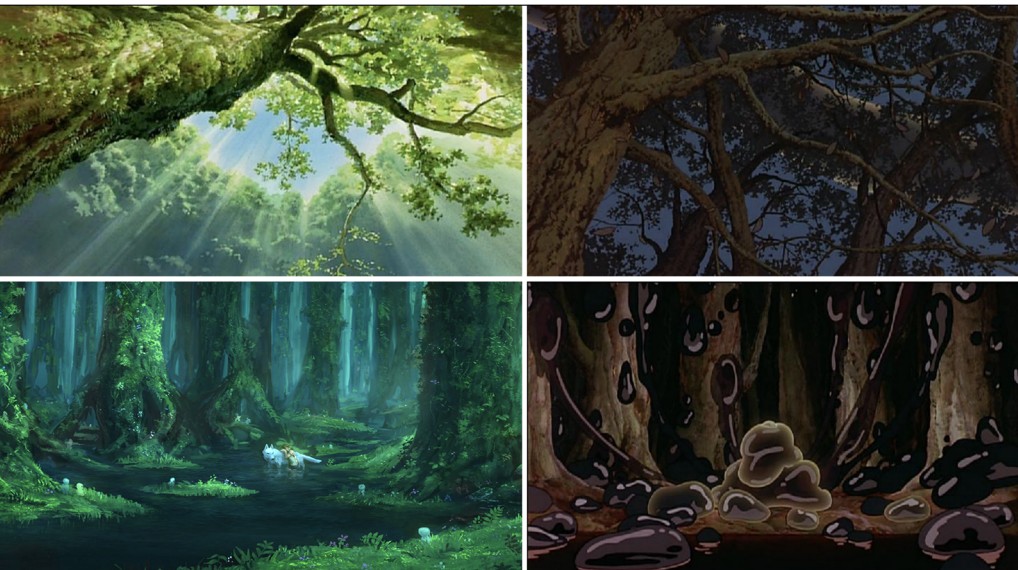

**Figure 3.** Screenshots from the "Princess Mononoke" film. Images processed by authors.

This dynamic representation of the landscape contributes to creating a sense of realism and vitality in the films. In most of Miyazaki's films, there are detailed and evocative depictions of seasonal landscapes, such as forests draped in golden autumn hues, flower-filled meadows in spring, or snow-capped mountains in winter. These different seasons not only enrich the visual beauty of the scenes but also hold symbolic significance. Seasons represent the passage of time, the cycle of life and can evoke a range of emotions and themes, such as rebirth, transition or nostalgia.

Furthermore, Miyazaki's films often feature fantastical and surreal landscapes that blend natural and invented elements [6]. For instance, "Howl's Moving Castle" presents a landscape of floating cities in the sky (Figure 4) [7], while "Spirited Away" showcases a world where nature merges with ancient and magical human structures.

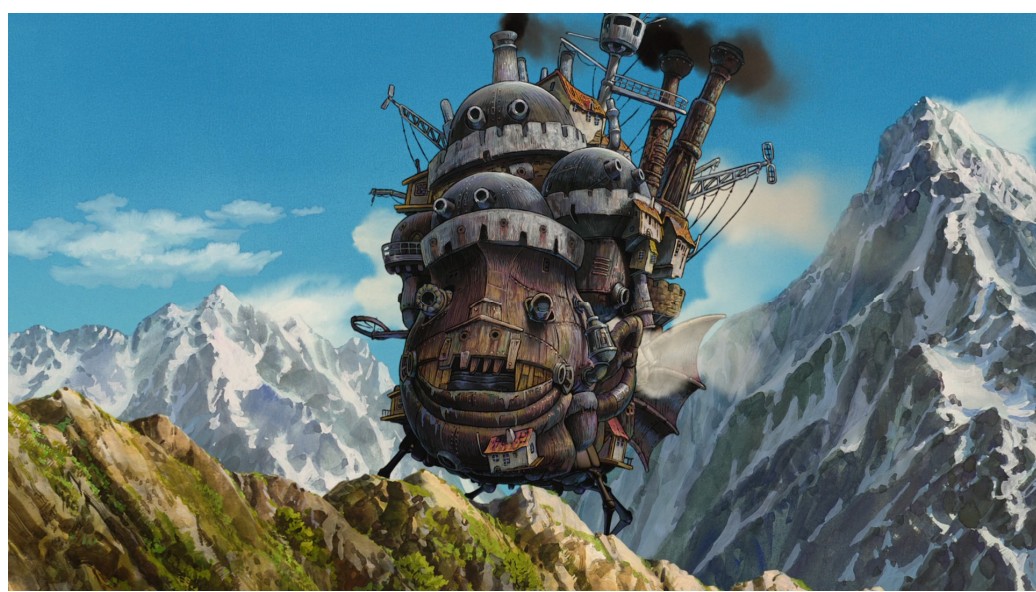

**Figure 4.** Screenshot from the "Howl's Moving Castle" movie. Image processed by authors.

These fantastic landscapes provide an escape from reality and offer a unique visual experience that stimulates the viewer's imagination and creativity.

Finally, a character-defining element in the representation of landscapes in Miyazaki's films is the depiction of spirits, magical creatures and supernatural elements that provide landscapes with a spiritual and mystical dimension, highlighting the profound interconnection between the natural and supernatural realms. For example, in "Princess Mononoke", the forest is depicted as a sacred place inhabited by animal spirits and deities (Figure 5). This portrayal conveys respect and a sense of wonder for the natural world, encouraging a broader and deeper understanding of reality. Overall, the representation of landscape in Miyazaki's films goes beyond mere aesthetics and becomes a powerful narrative and symbolic tool. Through an attention to detail, the temporal dimension, fantastical landscapes and spirituality, Miyazaki invites us to reflect on the beauty and importance of nature [8].

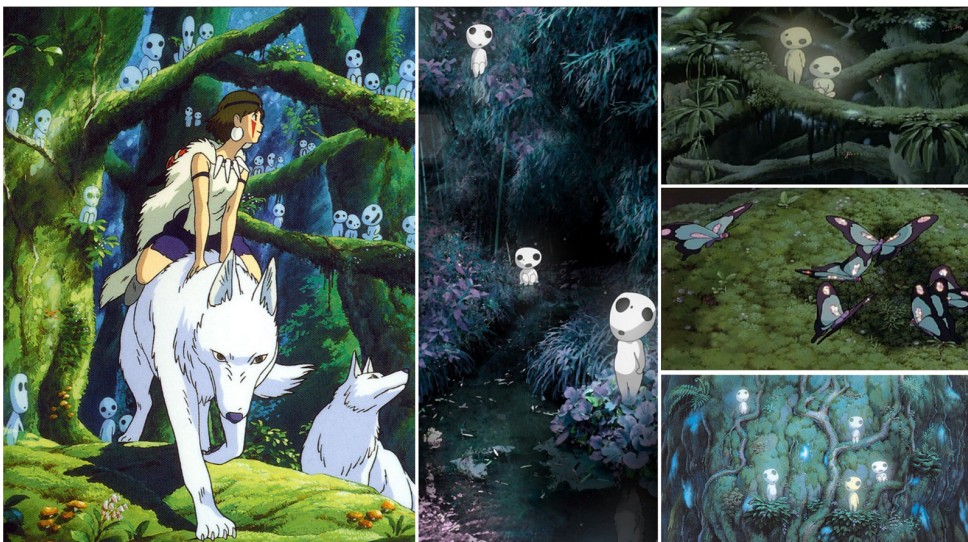

**Figure 5.** Screenshots from the "Princess Mononoke" film (1997), showing different scenes from Shishi Gamiís Forest depicting Kodoma spirits and butterflies in the forest. Images processed by authors.

## 2. Ecology in Miyazaki's Films Production

Our research stressed the importance of landscape depictions in Miyazaki's filmography and its role in increasing and developing environmental awareness. Miyazaki has an ecological vision that can be deduced in his filmography. Every element of the landscape is depicted highlighting the beauty and complexity of natural elements. Landscape representations communicate the importance of preserving biodiversity and the delicate ecosystem balance. This distinctive feature of Miyazaki's aesthetics is expressed through the use of drawing. In "Heidi, Girl of the Alps" (1974), Miyazaki represented the mountain landscape through evocative natural environments, reproducing the atmosphere of the Swiss Alps where the story is set (Figure 6).

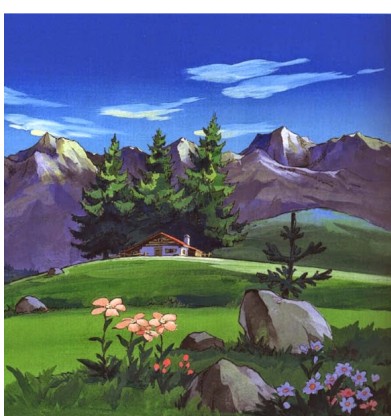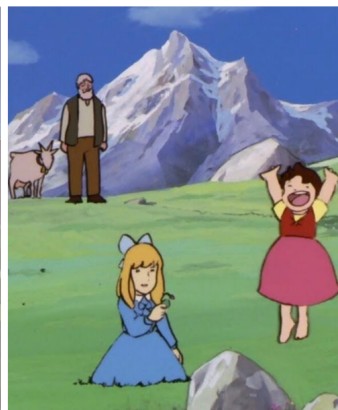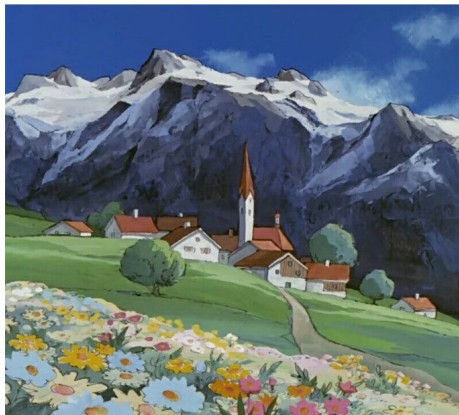

**Figure 6.** Screenshots from the "Heidi, Girl of the Alps" film. Images processed by authors.

The settings are artistically rendered, creating an engaging and suggestive environment that seamlessly blends with the story and the emotions of the characters. His ability to create such detailed and evocative images through freehand drawing is connected to his early experience as a designer. Miyazaki's attention to detail and his artistic mastery are manifested in every aspect of the landscapes he creates. Every leaf, every flower and every ray of light is meticulously drawn to capture the beauty and vitality of the natural world. This level of detail enriches the viewers' visual experience, creating a sense of immersion in the depicted world. The precise lines and attention to detail enhance the beauty of the landscapes and create an emotional connection with the viewers. Drawing becomes a means to convey the essence and atmosphere of the depicted places, offering an engaging visual experience. The use of lines is particularly distinctive in Miyazaki's landscape drawings. The fluid and organic lines create a sense of movement and vitality in the landscapes, evoking a feeling of flow and harmony with the surrounding environment [9].

Lines can also be used to define architectural forms, creating detailed and realistic urban landscapes that seamlessly integrate into the fantastical worlds of his films. In the film "Porco Rosso", through the use of vibrant colors, meticulous details and evocative atmospheres, the landscape contributes to creating an immersive and engaging world that harmoniously blends with the film's plot and characters (Figure 7).

The interaction between landscapes and characters is another important aspect of Miyazaki's works. Landscapes in Miyazaki's films take on an emotional, metaphorical dimension. They become symbols and visual representations of the characters' emotions and inner experiences [10]. For example, a stormy landscape may reflect a character's inner turmoil, while a serene and luminous landscape can symbolize hope and healing. Landscapes, by influencing emotions, choices and character developments, become a central element for characterization and storytelling in Miyazaki's films. Characters often find inspiration and wisdom from the surrounding nature, showcasing a deep connection between humans and their natural environment.

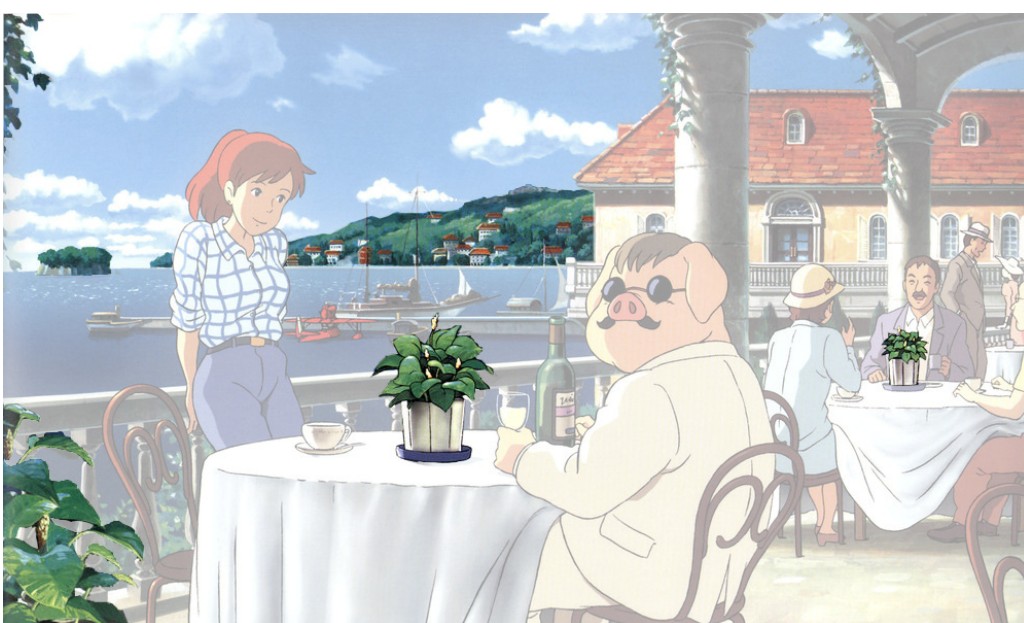

**Figure 7.** Screenshot from the "Porco Rosso" film. Image processed by authors.

In "Howl's Moving Castle", the magical landscape of the wandering castle is intrinsically linked to the characters. The castle itself transforms and adapts to the emotions and challenges of the protagonists, reflecting their experiences and emotional growth. The vast skies and spectacular clouds in "Howl's Moving Castle" convey a sense of freedom and adventure [11]. In "Nausicaä of the Valley of the Wind", the toxic deserts express a sense of desolation and threat. Landscapes, therefore, become a powerful tool to emotionally engage the audience and fully immerse them in the film's world.

Miyazaki uses landscapes as symbols and metaphors to explore deeper themes. For example, the sea present in many of his films represents both an element of danger and endless possibilities. In the film "Spirited Away", the floating city becomes a symbol of hope and redemption. Another recurring theme is the duality between urban and rural landscapes. While rural landscapes are often portrayed as places of magic, spirituality and harmony with nature, urban landscapes are depicted as frantic, polluted and alienating. This contrast highlights the tension between technological progress and the loss of connection with nature, stimulating critical reflection on the direction of humanity and the impact of individual and collective choices on the environment [12]. In the film "Kiki's Delivery Service", a clear duality between urban and rural landscapes is presented. The bustling city where Kiki carries out her work contrasts with the tranquility of the coastal village, creating tension between the hectic city life and the simplicity of rural life.

In a nutshell, the analysis of landscapes in Hayao Miyazaki's films reveals the central importance of these elements in creating a unique cinematic experience. The ecological vision, the interaction between landscape and characters, the duality between urban and rural landscapes, the spirituality of landscapes, the attention to detail and the artistic mastery all come together to convey profound messages about ecological balance, connection with nature and the emotional richness of the natural world. Miyazaki invites us to reflect on our relationship with the environment and our responsibility to preserve it for future generations [13]. His films show us that the landscape is not just a visual backdrop but a vital protagonist that contributes to understanding and appreciating the world around us. Miyazaki's films are not slogans against ecology or warnings of the end of the world. Instead, they are moments that seek to make us appreciate a natural world that may no longer surround us. The curiosity of his characters ignites the inner spirit of each of us, inspiring us to venture out and explore [14].

## 3. Materials and Methods

The methodological approach adopted combines critical viewing of the films with research into Miyazaki's creative process and influences. The present study proposes a critical analysis of a selection of Miyazaki's animated films, chosen on the basis of visual themes rather than chronology. Eleven films were identified as representative of the variety of landscapes represented in his work (Figure 8). These include: "Princess Mononoke", "My Neighbor Totoro", "Howl's Moving Castle", "Spirited Away", "Ponyo on the Cliff by the Sea", "Nausicaä of the Valley of the Wind", "Kiki's Delivery Service", "From Up on Poppy Hill", "Castle in the Sky", "The Wind Rises" and "Porco Rosso". The above films were selected on the basis of the following criteria:

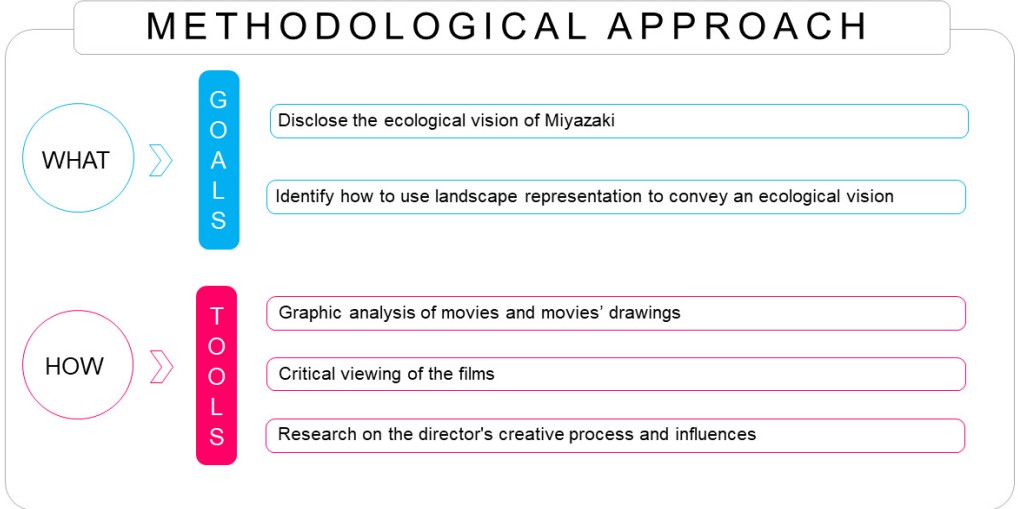

**Figure 8.** Diagram of the research methodology. Source: authors.

- The presence of landscape (either natural or man-made) in all film settings;
- The direct or indirect presence of an ecological perspective at the narrative, descriptive and informative levels;
- The dichotomy between natural and human elements;
- The use of immediate, symbolic and metaphorical images;
- Miyazaki's desire to elevate video animation from a minor art form to an aesthetic code for adults;
- The density of the films and the space left for interpretation, in order to continue to appreciate the images and their meanings after the film has been seen.

In terms of materials, to better explain the choices made for the selected films, a short synopsis of the selected films is provided.

"My Neighbor Totoro": Set in the Japanese countryside in the 1950s, the film follows the adventures of two sisters, Satsuki and Mei, who move to a rural house to be closer to the hospital where their mother is staying. The landscape surrounding the house is lush and enchanting countryside, with forests, hills and green fields. The trees are depicted with great artistry and are filled with details, while rivers and lakes are rendered realistically. The landscape becomes a place of exploration and adventure for the two sisters, as well as a refuge of tranquility and comfort during difficult times [15] (Figure 9).

"Princess Mononoke": Set in a fantasy world, the film features a vast forest populated by nature spirits and mythical creatures. The imagery of the forest is spectacular, with trees towering against the sky and abundant flora. The forest is painted with vivid and nuanced colors, creating a sense of magic and wonder [16].

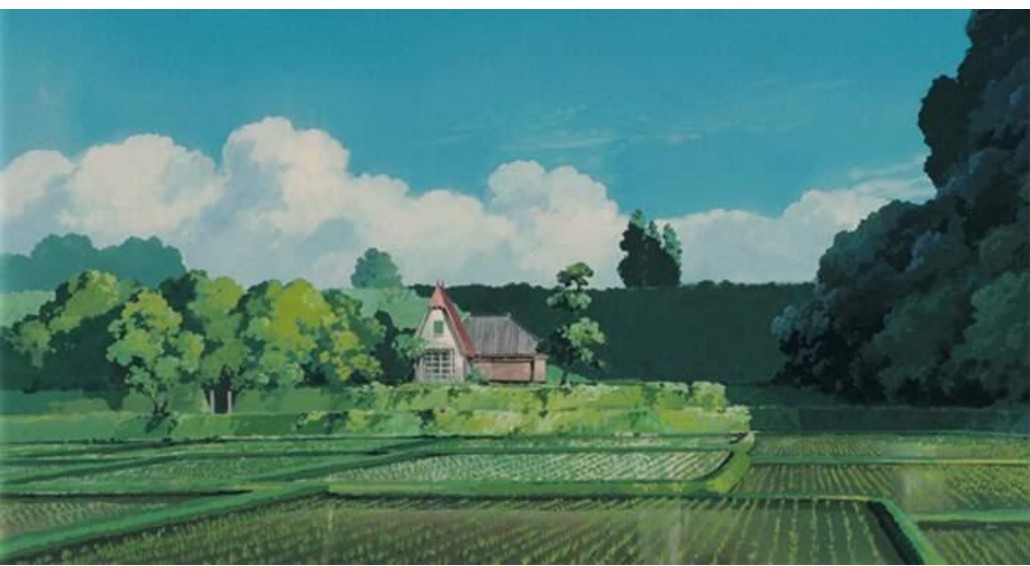

**Figure 9.** Screenshot from the "My Neighbor Totoro" film. Image processed by authors.

"Howl's Moving Castle": The film follows the adventures of Sophie, a young girl transformed into an old woman by a curse, and her encounter with the mysterious wizard, Howl. The landscape in the film is characterized by a fairy-tale atmosphere and a variety of evocative settings. From bustling and noisy towns to green and idyllic countryside, to Howl's majestic moving castle traversing the lands. The landscapes include lush and vibrant meadows with colorful flowers and rolling hills [17]. There are also scenes depicting busy and bustling cities, with intricately designed and detailed architecture (Figure 10).

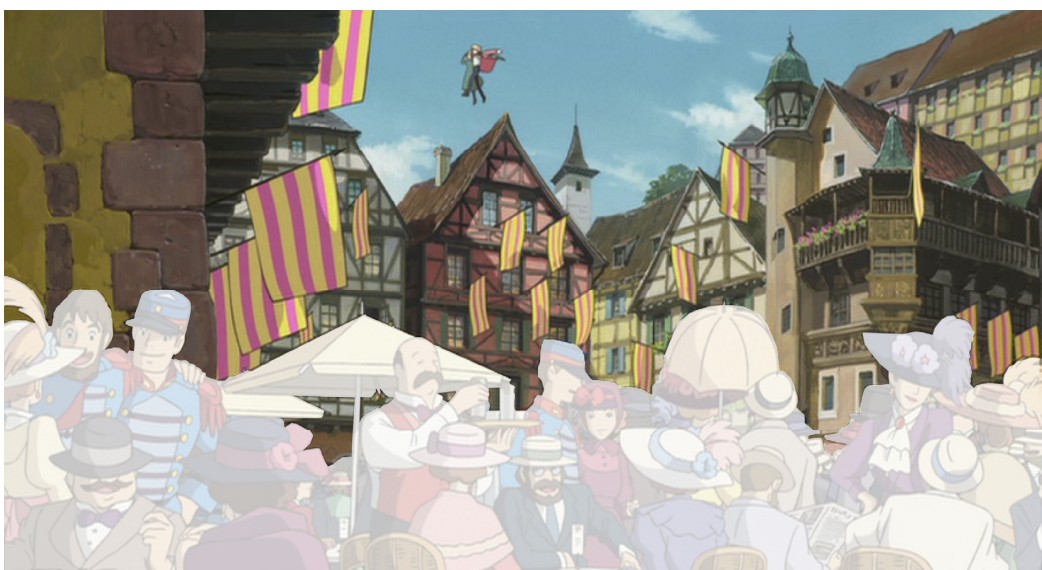

**Figure 10.** Screenshot from the "Howl's Moving Castle" film. Image processed by authors.

"Ponyo on the Cliff by the Sea": Set in a coastal village in Japan, the film tells the story of Ponyo, a little fish-girl who wishes to become human and live on land. The marine landscape in the film is rendered with a meticulous attention to the colors and forms of marine creatures, creating a captivating and immersive representation of the ocean. The cliffs, beaches, waves and marine inhabitants are recreated with great artistic skill, conveying a sense of wonder and connection with nature. The cliff becomes a landmark and a meeting place for the characters, representing the strength and beauty of the surrounding nature. The film showcases a picturesque coastline with crystal-clear waters and fascinat-

ing cliffs [18]. The underwater scenes are particularly beautiful, with colorful corals and a variety of fascinating marine creatures. The colors and light of the ocean are rendered vibrantly and enchantingly.

"Spirited Away": Set in a spiritual dimension called "Spirited Away", the film tells the story of Chihiro, a young girl who stumbles upon a world inhabited by spirits and mystical creatures. This film features an imaginary city with a unique atmosphere [19]. Urban landscapes are depicted with great attention to architectural details, combining traditional Japanese elements with a touch of futurism. The nighttime scenes of the city, with lantern lights and reflections on the water, create a magical atmosphere. The landscape conveys a sense of mystery and abandonment.

"From Up on Poppy Hill": Set in 1960s Yokohama, the film follows the story of Umi Matsuzaki, a young girl who manages a family boarding house situated on a hill overlooking the harbor. The maritime and harbor landscape of Yokohama is depicted in a detailed and captivating manner in the film. The choice of Yokohama as the setting allows for the exploration of themes of tradition and progress, the beauty of the sea and the challenge of preserving cultural identity in a context of rapid urban development. The rapidly changing city and the enchanting view of the harbor provide an evocative backdrop for Umi and her friends' story. The surrounding natural landscape, with its green hills, fields of poppies and the sea extending to the horizon, represents an homage to the beauty and fragility of nature.

"The Wind Rises": Set in the first half of the 20th century, the film tells the story of Jiro Horikoshi, a Japanese aeronautical engineer who contributed to the design of military aircraft during World War II. The landscape in the film spans from the Japanese countryside to rapidly developing cities, providing a visual context for Jiro's story. The rural countryside evokes a sense of peace and connection with nature, while the transforming cities represent the rise of industrialization and the frenzy of technological progress [20]. The wind, constantly blowing in the film, becomes a symbolic presence in the landscape. It represents the strength and energy of nature, as well as the change and transience of life. The wind plays a metaphorical role in the film, reflecting Jiro's passion for flight and his ambition to create innovative airplanes despite challenges and difficulties.

"Nausicaä of the Valley of the Wind": Set in a distant future, the film tells the story of Nausicaä, princess of the Valley of the Wind, who strives to protect nature and seek peace in a world divided by conflicts and devastation [21]. The landscape in the film reflects the impact of human actions on the environment. The film's landscape is also characterized by extraordinary and unique flora and fauna. Mutant creatures and giant insects are integral parts of the landscape, representing a form of life adapted to extreme and ever-changing conditions. This aspect emphasizes nature's ability to regenerate and find balance even in extreme situations.

"Kiki's Delivery Service": The film follows the adventures of Kiki, a young witch who decides to settle in a coastal city for a year as part of her family tradition. Using her ability to fly on a magical broomstick, Kiki opens a delivery service, coming into contact with different characters and places in the city [22]. The landscape plays a significant role in evoking a magical atmosphere and depicting the growth and self-affirmation journey of the young witch protagonist. The urban and natural landscapes intertwine to create an enchanting environment that captivates the audience and provides an emotional backdrop for Kiki's story.

"Castle in the Sky": This film tells the story of Sheeta, a young girl on the run who possesses a mysterious medallion that connects her to the legendary floating city of Laputa, hidden among the clouds. Sheeta is rescued by Pazu, a young boy who dreams of reaching Laputa and uncovering its secrets [23]. The landscape takes on symbolic meaning, representing the connection between humans and nature, as well as the importance of preserving the balance between them. The flying castle, with its destructive power and potential for evil exploitation, highlights the theme of the irresponsible use of natural resources and technology. Further, the living spaces depicted represent materials, shapes

and forms linked to the tradition of the places where they are located. In this sense, the author distances himself from the 'Metabolist' movement, which aims at the extreme optimisation of rapidly changing spaces and shows how the serenity of a life anchored in tradition and on a human scale is still possible.

"Porco Rosso": The story takes place in a Mediterranean-inspired setting during the 1930s, and the landscape drawings faithfully reflect the locations and atmospheres of that era [24]. The original idea for the film dates back to 1984, when it was a simple comic book. Miyazaki later became passionate about the story and decided to make a medium-length film to be shown on Japan Airlines planes (which sponsored the production), but the project then expanded into a 93-minute film that was completed in 1992. The landscape drawings are rich in details and shades, with particular attention being given to the depiction of different seaside locations, islands and coastal cities (Figure 6). Crystal-clear waters, beaches, ports and buildings are represented with great precision and style, offering a fascinating visual spectacle. The film contrasts the peaceful beauty of Mediterranean architecture with the shadow of war that haunts the film's protagonist, Marco Pagot, a former Royal Air Force ace disguised as a pig.

To analyze the landscapes, we carried out detailed visual observations of the films, taking note of various landscape elements such as topography, geology, hydrology, flora and fauna, anthropic elements, atmosphere and lighting (Figure 11). We paid particular attention to details, color nuances, lines and shapes that constituted the landscapes, aiming to grasp Miyazaki's artistic intentions and the emotional effect that the landscapes conveyed.

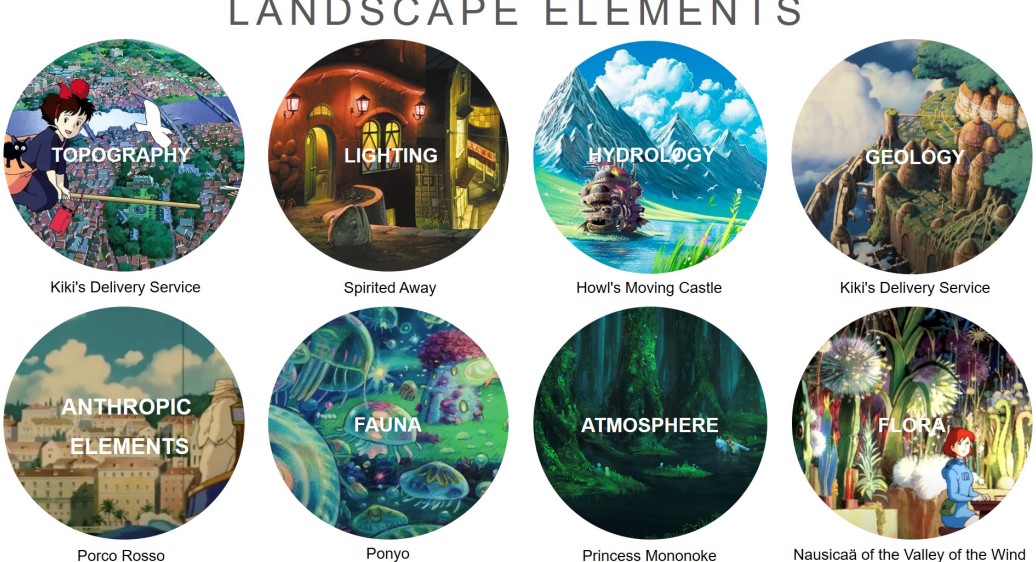

**Figure 11.** Character-defining elements of the landscape representations in Miyazaki's filmography. Images processed by authors.

Furthermore, we carried out in-depth research on works written about Hayao Miyazaki, in order to better understand his ecological vision and the influences that guide his representations. This allowed us to contextualize Miyazaki's creative choices and explore the connections between his imaginary world and his personal beliefs regarding ecological balance and nature conservation. This critical perspective provided us with additional insights and helped us delve deeper into the role of landscapes in conveying ecological themes through storytelling and communication.

Finally, we organized our observations and analyses coherently, identifying recurring themes and motifs that arise from the landscape representations in Miyazaki's films. Through this combined methodology of visual observation, source research, critical analysis and the contextualization of Miyazaki's works, we were able to thoroughly explore his strategy to disseminate and communicate his ecological vision.

## 4. The Creative Process: From Drawing to Video Animation

In terms of the creative process and the techniques used, this study examined the role of drawing in the representation of landscapes in Miyazaki's films. The artist and his team employed a variety of techniques and tools to transform their vision into detailed and evocative images.

First and foremost, hand-drawn sketches were a fundamental part of the creative process. The artists at Studio Ghibli use pencils, crayons and pastels to create their initial sketches and illustrations. In some cases, Miyazaki personally handles the landscape drawings, as his artistic skill and vision are essential to achieving the desired result. His initial sketches evolve into authentic works of art, with each element carefully designed to convey a specific atmosphere and evoke precise emotions in the viewer (Figure 12).

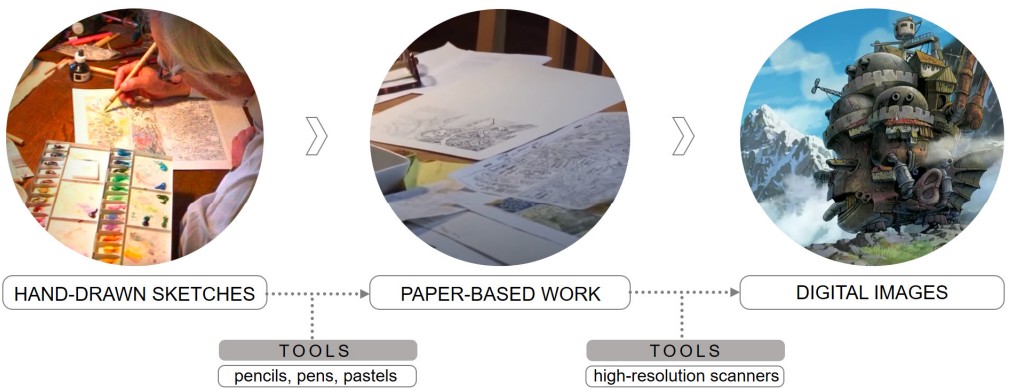

**Figure 12.** The creative process of Miyazaki and his team. Image edited by the authors.

His mastery of drawing and deep understanding of landscapes enable him to create scenes that convey emotion, atmosphere and meaning. The traditional tools he uses in the early stages of the creative process allow him to explore the shapes, details and lighting effects of the landscapes to be animated. Once the basic drawings have been created, the animation team use high-resolution scanners to convert the paper-based work into digital images. This step preserves and refines the details of the drawings and provides a digital base for further processing. Artists then use a variety of coloring techniques to bring life and depth to the landscapes. A common technique used is watercolor painting, which allows for subtle and gradual tonal variations, giving the landscapes a harmonious and natural appearance.

Digital tools such as graphic tablets and video animation software are then used to add detail, shading and special effects. Miyazaki skillfully uses color to create atmosphere and evoke emotion. His landscapes range from vibrant and bright to dark and gloomy, depending on the narrative context. His skillful use of tone and contrast helps to convey feelings of wonder, mystery or suspense, enriching the overall visual experience. The artists work closely with production designers, animators and lighting technicians to ensure that the landscapes come to life in accordance with Miyazaki's artistic vision. Lighting plays a fundamental role in emphasizing the details, shadows and atmosphere of the landscapes, helping to create an evocative and realistic environment. The skillful use of perspective, light, shadow and architectural detail helps to create believable and engaging landscapes.

Miyazaki's choice to begin the creative workflow with traditional illustration techniques and analogic drawing gives him greater flexibility and makes it easier to manage the complexity of his animations.

For example, in the case of the film "Spirited Away", the Japanese animator's intention in the graphic composition of the environments of the animated baths is not to give life to a parallel spiritual universe, but to evoke a mental dimension, somewhere between dream and memory, to reactivate a lost gaze, to which he gives an aesthetic staging. Thinking of the image as a mental image, or rather an interior image, rather a well-conceived

and functioning prototype of the real world, allows Miyazaki to combine characters and places in a symbolic manner, and to manipulate shapes and colors in order to give life to scenarios that activate the imagination without any rational pretence [25].

Diametrically opposed to Disney, whose contemporary concern with mimesis gives images an almost photographic value in relation to the real world, Studio Ghibli's operation here stages a structurally non-real place where each element is activated according to its own aesthetic regime. For example, the simplicity of Chihiro's and Aku's facial features, as opposed to the amount of detail in Yubaba's face, invites the viewer to read a conceptual clue into the film, and shows how 2D representations succeed in restoring the inexhaustible potential of the imagination, resisting the temptation to obey the principle of non-contradiction, to create imaginary and ideal creatures from scratch, rather than copies of real creatures.

The drawings of "Spirited Away" are deliberately non-realistic, with a delicate aesthetic that, more than 20 years later, remains an artistic model to aspire to when overstimulated by three-dimensional and immersive visualizations. Miyazaki's visual language tells of a contemplative space in which the "enchantment" that animates the city is a departure from contemporary reality.

With regard to the representation of man-made and natural landscapes, some general considerations concern the importance of the direct observation of nature. Miyazaki and his team at Studio Ghibli explored real landscapes as a source of inspiration. The artists visited the natural locations, took photographs and studied them, gathering details through direct observation. This careful observation and direct contact with nature allows the artists to capture its beauty and complexity, which is then reflected in their drawings and final works (Figure 13).

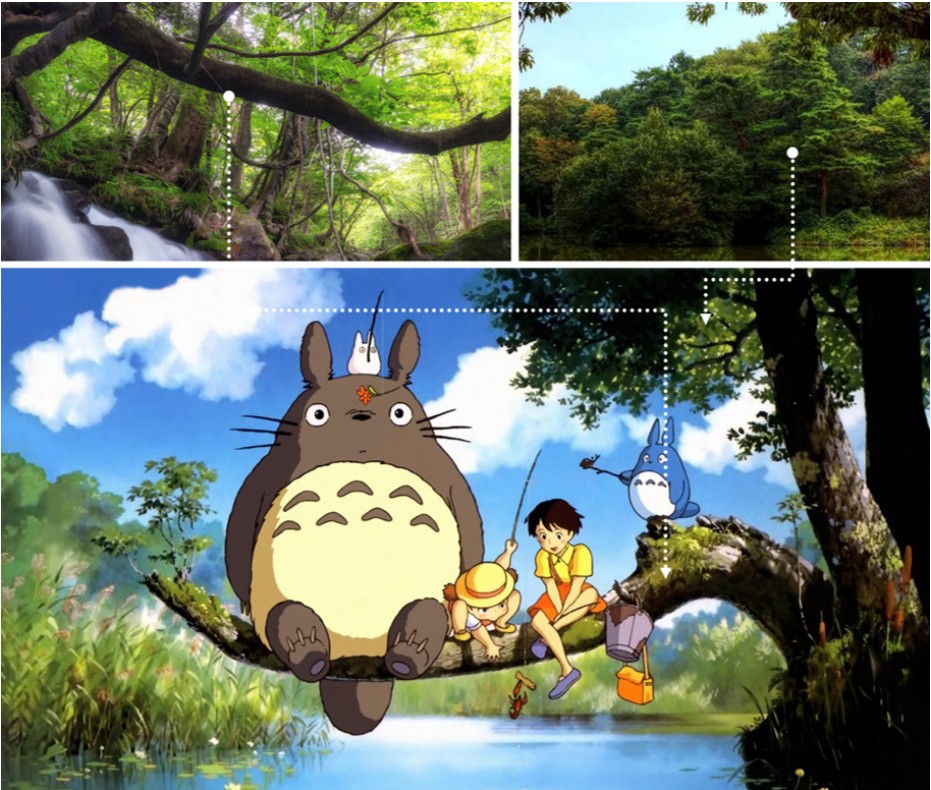

**Figure 13.** Example of the role of direct and indirect observation of the natural environment in the representation of landscape in Miyazaki's filmography. Image edited by D. Mezzino.

In "Porco Rosso", for example, the imaginative Hotel Adriano is located on an islet that can be identified as the islet of San Giovanni in Lake Maggiore (current district of Verbania, Italy) (Figure 14).

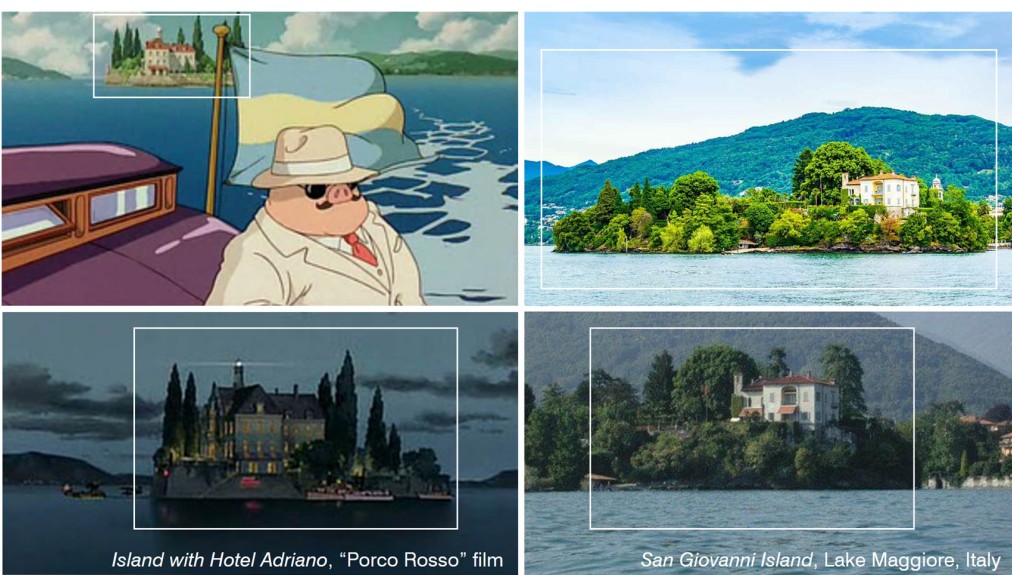

**Figure 14.** Example of the role of direct and indirect observation of the natural and built environment in the representation of the landscape in the film "Porco Rosso". Image edited by D. Mezzino.

Another example of the importance of the direct observation of the landscape (in this case, the urban landscape) in Miyazaki's creative process is the depiction of the city of Milan in the film "Porco Rosso". The Japanese director imagines Milan's Navigli as a system in which seaplanes can glide and take off at will. Miyazaki's drawings, however, show waterways and bridges inspired more by the Po in Turin than the Navigli in Milan. Comparing a view of the Ponte Vittorio Emanuele bridge with stills from the film, there are many similarities in image and architectural style (Figure 15). Furthermore, a drawing of the Mole Antonelliana appears between the film's credits, which could confirm the 'fusion' of elements from the cities of Milan and Turin.

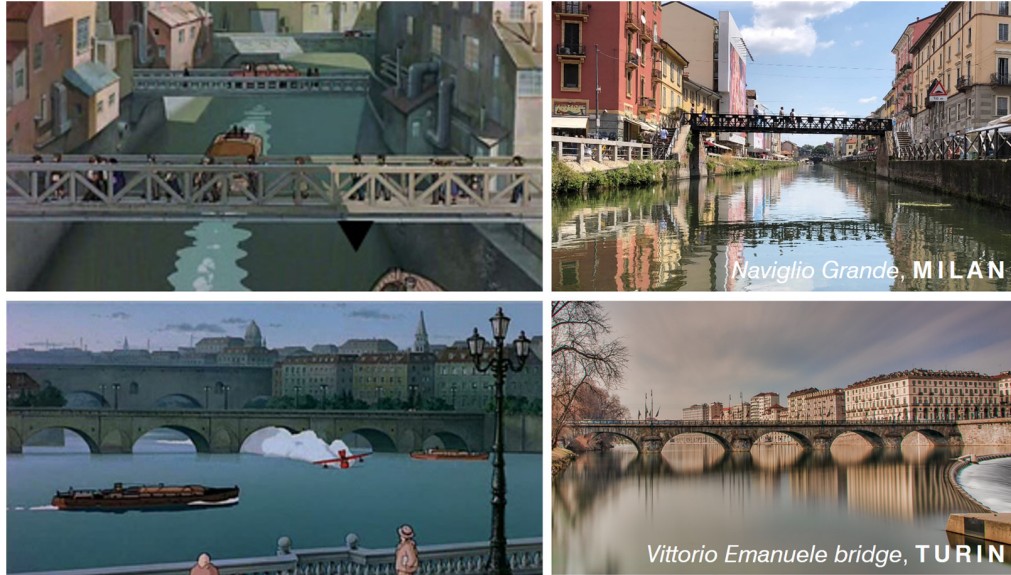

**Figure 15.** Example of the influence of direct and indirect observation of the urban landscapes of Milan and Turin on the representation of Milan in the film "Porco Rosso". Image edited by D. Mezzino.

### 5. Results: Visual Poetry for Environmental Awareness

Our analysis of the visual language of the eleven selected films showed the aesthetic value as well as the strong educational and poetic significance in conveying the concept and meaning of landscape. In Miyazaki's films, landscapes go far beyond mere backgrounds. The topography and geology of the worlds he creates often feature extraordinary formations such as towering mountains, mysterious canyons and islands floating in the sky. These geographical elements convey a sense of grandeur and adventure, suggesting a deep connection between the landscape and the main characters of the stories.

Hydrology is an important element, with watercourses such as rivers, streams and waterfalls accurately depicted, emphasizing the importance of water in natural life. Water plays an important role in Miyazaki's landscapes, symbolizing both the beauty and power of nature. Rivers, lakes and oceans are present in his films, conveying a sense of life and movement, and often revealing secrets and mysterious stories. This approach highlights the importance of water as a vital element and a force that shapes both the landscape and human communities.

Flora and fauna are depicted with great attention to detail. The enchanting creatures and magical plants of his imaginary worlds intermingle with real species of insects, birds and plants. This wealth of biodiversity underscores Miyazaki's love of nature and the need to preserve the diversity of species and ecosystems that enrich our planet (Figure 11).

In addition to natural landscapes, Miyazaki also depicts anthropic landscapes, such as cities, villages and human architecture. These urban landscapes often contrast with the surrounding nature, creating a duality that highlights the impact of human activity on the environment. This representation encourages reflection on sustainability and human responsibility towards landscapes and ecosystems [26].

Miyazaki's artistic mastery of landscape representation is also evident in his attention to detail. This study shows the attention to detail in the preparatory analogue drawings, which are fundamental to the animation created later. Many of these drawings were made directly by the masters, Miyazaki and Takahata, in order to make the work to be done as clear as possible to all the collaborators. By breaking down the scenes of each of the films analyzed, it was possible to understand the authors' methods and working mechanisms: from still backgrounds to moving objects, to the number of frames required for different types of scene. Landscape depictions are characterized by a wealth of visual details, from the structure of rocks to the color shades of the skies. This level of detail conveys a sense of realism and depth, engaging viewers in an immersive visual experience. The film "Ponyo on the Cliff by the Sea" is an example of Miyazaki's extraordinary attention to detail in marine landscapes. The use of colors, the shades of water and the fluid movements of marine creatures showcase his artistic mastery in realistically and captivatingly rendering marine landscapes.

Another notable aspect is Miyazaki's ability to use landscape as an emotional metaphor. Through the creative use of landscapes, Miyazaki conveys deep moods and feelings. An example is the dark and haunting landscape in "Spirited Away", which reflects the evil spirit that threatens the city itself. In the film "From Up on Poppy Hill", the rapidly transforming urban landscape serves as an emotional metaphor for loss and change. Images of old houses and construction sites evoke a sense of nostalgia and the struggle to preserve the past.

The analysis of Hayao Miyazaki's films conveys profound messages about the importance of connecting with nature and our role in the ecological balance. His ability to combine fantastical and realistic elements creates a captivating visual and emotional experience for audiences. The results of an in-depth analysis of landscapes in Hayao Miyazaki's films are manifold and significant. Some key points are summarized below.

■   *Environmental awareness:* Miyazaki's films highlight the importance of environmental conservation and sustainability. Through attention to detail and the use of complex and diverse natural landscapes, the author invites us to reflect on our impact on the environment and the need to preserve biodiversity and ecosystems. "Princess

Mononoke" is one of the most emblematic films of Miyazaki's ecological vision. The landscapes of dense and untouched forests are depicted with great attention to detail, creating a vibrant natural environment where humans, animals and spirits coexist and influence each other [27].

- *Connection with nature:* The landscapes depicted in Miyazaki's films invite us to rediscover the magnificence and significance of nature in our existence. They remind us of our deep interconnectedness with the natural elements and implore us to foster a deeper and more reverent relationship with the environments that surround us.
- *The power of imagination:* The fantastic landscapes created by Miyazaki show us the power of imagination and creativity in shaping the world around us. They encourage us to look beyond appearances and explore new possibilities, prompting us to cultivate an open and curious mind.
- *The spiritual essence of nature:* The landscapes in Miyazaki's films often take on a spiritual dimension, reinforcing the idea that nature and the surrounding environment can have a profound influence on our inner lives. They invite us to recognize the sacredness and beauty of nature and to develop greater spiritual awareness. In the film "My Neighbor Totoro", the countryside house surrounded by a dense forest becomes a meeting place between the human world and the realm of spirits. These spiritual landscapes reveal Miyazaki's vision of the interconnectedness of the visible and the invisible, and his reflection on the role of humans in the grand scheme of the universe. Majestic trees and natural environments create a sense of wonder and connection with nature, embodying Japanese spirituality, which is associated with trees and fantastic creature spirits. In the film "Spirited Away", the landscape represents a deep connection between nature and the supernatural. Lush forests, crystal-clear rivers and imposing mountains are inhabited by spirits and magical creatures, who seek refuge in these places. The connection between nature and the spiritual realm emphasizes the significance of preserving the natural environment and recognizing the intrinsic bond between humanity and nature itself.
- *The graphic language:* The meticulous attention to detail and artistic expertise in depicting landscapes underscores the role of drawing as a powerful means of conveying complex messages.
- *Critical reflection on urban dynamics:* The juxtaposition of urban and rural landscapes in Miyazaki's films stimulates a critical reflection on society, highlighting the contrasting dynamics between them and encouraging us to consider the social and economic factors that influence both the built and natural environments. It encourages us to question the dominant model of development and to consider the impact of our choices on the natural world [28].

This analysis of the representation of landscape in Hayao Miyazaki's films invites us to rethink our relationship with nature, to value the art of filmmaking as a powerful means of expression and education, and to encourage a critical reflection on society and the human impact on the environment.

In addition, the focus of this research is to understand the approach, principles and workflows of Miyazaki's visual language in order to reproduce it in contemporary video animation, with the aim of raising awareness of the landscape and stimulating critical thinking about balanced growth, including natural and anthropic needs.

## 6. Discussion: Reflections on Landscape Representation Technique

This study opened the discussion on seven main themes related to landscape.

*The role of landscape in storytelling:* One of the noteworthy aspects of Miyazaki's films is the role of landscape as a crucial narrative element [29]. For instance, within his films, the depiction of an enchanted forest serves to represent a realm of adventure and personal exploration, whereas a chaotic urban environment can symbolize a fast-paced and alienating society. In "The Wind Rises", the evocative imagery of airplanes soaring through the expansive sky and the undulating countryside evokes a feeling of liberation, effectively

conveying the dreams and aspirations of the protagonist, Jiro Horikoshi. Similarly, in "Spirited Away", the urban landscape of Japan becomes a suffocating and isolating society, portrayed through bustling streets and towering skyscrapers. This starkly contrasts the dreamlike setting of the underground city and the lush natural scenery, which symbolize hope and rebirth. Lastly, in "My Neighbor Totoro", the rural landscape, featuring elements such as the colossal cedar tree and the flowing stream, serves as a backdrop for adventure and enchantment. These locations act as a bridge between reality and fantasy, driving the plot forward and accentuating a sense of awe and discovery [30].

*Landscape as cultural expression:* The landscapes depicted in Miyazaki's films intricately portray the complex interplay between Japanese culture and the natural environment. They serve as a canvas for showcasing the Japanese tradition of appreciating subtle beauty and cultivating a harmonious relationship with nature. These landscapes are meticulously crafted, paying attention to minute details that reflect these cultural values [31]. For instance, the presence of Zen gardens, temples and traditional countryside houses in numerous scenes highlights the significance of spirituality and the pursuit of harmony with the natural world in Japanese culture. These elements serve as visual cues, emphasizing the deep-rooted connection between the Japanese people and their environment. In the film "Princess Mononoke", the landscape of the Forest of the Spirits and the surrounding mountains embodies the profound relationship between nature and spirituality within Japanese culture. Sacred sites like the Spirit's Well and ancient trees evoke a sense of reverence and a profound connection with the forces of nature. Likewise, in "Ponyo on the Cliff by the Sea", the marine landscape and its inhabitants become vessels for Japanese myth and tradition. The film explores the concept of the sea as a living entity, teeming with magical and mysterious creatures, drawing inspiration from Japanese cultural beliefs and folklore. Through these deliberate portrayals, Miyazaki's films pay homage to the intricate bond between Japanese culture and the natural world. The landscapes serve as a medium to capture the essence of Japanese identity, emphasizing their deep respect for nature and the interconnectedness of all things.

*Urban development and natural environment:* Miyazaki's films shed light on the detrimental effects of urban landscapes on both the environment and society [32]. They serve as a poignant reminder of the consequences of uncontrolled urbanization and unsustainable development practices [33]. In "Spirited Away", a grim and polluted metropolis is portrayed, where nature has been replaced by concrete structures and toxic emissions. This depiction prompts viewers to reflect on the choices made in urban development and the need for a harmonious balance between human needs and environmental preservation. Similarly, in "Howl's Moving Castle", the urban landscape of Ingary is characterized by a polluting black fog, symbolizing the corruption and environmental degradation caused by reckless resource exploitation. This portrayal highlights the significance of adopting sustainable approaches to urban development and the preservation of the natural environment. "Ponyo on the Cliff by the Sea" addresses the issue of marine pollution and its impact on marine life. The film portrays a polluted marine landscape and human waste infiltrating the sea, serving as a critique of environmental negligence and a reminder of the crucial need to protect marine ecosystems. In "Spirited Away", the landscape serves as a critique of human greed and the exploitative nature of profiting from the destruction of the environment. The presence of a residential spa for spirits, managed by dark and corrupt characters, symbolizes the ruthless exploitation of nature for personal gain. This theme underscores the importance of respecting and valuing nature while maintaining a sustainable equilibrium between human interests and environmental conservation. Overall, Miyazaki's films serve as powerful narratives that raise awareness about the negative impacts of unchecked urban landscapes, emphasizing the urgent need for responsible urban planning, environmental stewardship and the preservation of natural resources.

*The connection between landscape and human well-being:* The natural landscapes portrayed in Miyazaki's films evoke a sense of awe, inner tranquility and a profound connection with the natural world. They serve as integral elements in enhancing human

well-being and highlighting the significance of preserving these spaces for future generations. In "My Neighbor Totoro", idyllic and lush rural scenes act as a sanctuary for the protagonists, offering solace and assisting them in overcoming challenging circumstances. This raises important questions about the role of natural landscapes in promoting human well-being and emphasizes the need to protect and preserve these areas for the benefit of future generations. Similarly, in "Princess Mononoke", the pristine landscapes of forests and mountains are depicted as sanctuaries for the characters, providing them with peace and healing in the embrace of nature. This underscores the therapeutic power of natural landscapes for human well-being and highlights the importance of greater conservation efforts to safeguard these invaluable spaces. Miyazaki's films bring forth a profound understanding of the restorative qualities of natural landscapes, emphasizing their vital role in enhancing human well-being and promoting a deeper connection with the environment. They serve as a reminder of the need to value and preserve these spaces for the benefit of current and future generations.

*The impact of visual imagery on enhancing environmental consciousness:* Miyazaki's films serve as powerful examples of how art, particularly cinematic art, can effectively raise awareness and ignite interest in environmental conservation. Through their stunning and captivating depictions of natural landscapes, these films have the potential to inspire audiences, instilling a sense of wonder and a heightened concern for the environment. "Princess Mononoke" tackles complex environmental issues such as deforestation and the conflict between humans and nature. By shedding light on these important matters, the film prompts viewers to engage in critical reflection about their own relationship with the environment. It serves as a catalyst for raising awareness and encouraging a deeper understanding of the consequences of our actions on the natural world. Similarly, in "Nausicaä of the Valley of the Wind", the post-apocalyptic landscape of the Valley of the Wind, with its mutated flora and fauna, serves as a powerful representation of the devastating effects of war and environmental pollution. The film underscores the need for socially engaged art that not only entertains but also raises awareness of the environmental consequences of human actions. It emphasizes the responsibility of individuals and society at large to protect and preserve the environment. Moreover, the film's Taoist approach to life, agriculture and architecture, integrated into the natural world, clashes with the horrors of war; arising from the tragic real-world consequences of nuclear fallout, a horror will always remain in the background of Studio Ghibli's works. In the constant cycle of destruction and reconstruction, man appears from the outset almost alien to nature, thrown into contexts of industrialisation that humiliate him. This is the case for Pazu, who is shown at the beginning of the film at work in the coal mines, which are gloomy and cramped places on the inside, but with a fascinating exterior design, set in a natural context that fascinates the miners themselves.

These films demonstrate how the art of cinema can be a potent tool for promoting environmental consciousness. By presenting visually striking and thought-provoking images of natural landscapes, Miyazaki's films capture the audience's attention, fostering a connection to the environment and encouraging a sense of responsibility towards its preservation.

*Landscape as an economic and touristic resource:* The landscapes depicted in Miyazaki's films can be viewed as valuable economic resources and tourist magnets. A prime example is Jiufen, a town in Taiwan that served as inspiration for the enchanting landscapes in "Spirited Away". As a result of its association with the film, Jiufen has experienced a surge in tourism and has transformed into a highly sought-after destination. However, this situation prompts concerns about effectively managing tourism to strike a delicate balance between promoting these places and preserving their intrinsic integrity.

*The promotion of environmental consciousness and preservation:* The unspoiled and untouched natural landscapes portrayed in Miyazaki's films serve as catalysts for fostering environmental consciousness and preservation. The fascination evoked by these films can motivate efforts towards safeguarding natural areas and promoting sustainable practices. An illustrative case is the impact of "Princess Mononoke" on Yakushima National Park in Japan, where an upsurge in visitors occurred, driven by their desire to explore the land-

scapes that inspired the film. This phenomenon demonstrates how cinematic representations can effectively generate interest and appreciation for natural environments, leading to increased awareness and conservation initiatives.

Further research opportunities lie in investigating the artistic and cultural influences that have shaped Miyazaki's work. This could involve exploring Japanese artistic traditions like ukiyo-e (Japanese woodblock prints) and landscape painting, and their impact on Miyazaki's aesthetics and portrayal of landscapes [28]. Analyzing the literary, cultural and philosophical influences that have shaped his work can provide additional insights into the interplay between landscape and design in his films [29]. Delving into the technical aspects of drawing in Miyazaki's films could involve examining the materials and tools used by the director and his team, such as the choice of mediums like paper or canvas, and the use of techniques like ink, colored pencils and digital methods. Analyzing design techniques and production processes can offer a deeper understanding of artistic choices and the visual impact of the represented landscapes [30]. To explore the role of landscapes in Miyazaki's films as economic and tourist resources, research could focus on the socio-economic implications on the real locations that inspired the imaginary landscapes. This might involve interviews with local experts, analysis of tourism dynamics and land management policies, and investigations on the economic and social impact resulting from tourist interest generated by the films. Exploring these topics can deepen our understanding of the multifaceted nature of landscapes in Miyazaki's films and stimulate meaningful discussions on the relationships between humans, nature and artistic representation [31].

## 7. Conclusions

Our research illustrates the importance of landscape depictions in Miyazaki's filmography and its role in developing environmental awareness. This study demonstrates how Miyazaki's filmography is ideally placed to highlight and convey the essential components of what we call landscape, including environmental and cultural factors, and their interactions, in which visual perception plays a key role.

From a general perspective, the proposed study outlines how visual images serve as powerful narrative tools capable of conveying messages about the environment, culture and society. This study demonstrates how video animation can be used as a form of visual communication to stimulate environmental awareness. In addition, this study illustrates the role of traditional drawing and 2D representation in the video animation creation process, highlighting how it can convey mental images associated with landscapes and nature, indirectly stimulating their respect and preservation.

At a specific level, the adopted methodology enabled an analysis of various facets of the landscapes depicted in eleven selected films, uncovering essential insights into the role of drawing as both a form of representation and a tool for expressing Miyazaki's artistic vision. The research identifies the strategy and the character-defining elements adopted by Miyazaki in stimulating reflection on a sustainable combination of urban development and the preservation of natural elements, also increasing attention towards the landscape beauty, thus stressing the relevance of its preservation.

Furthermore, this study contributes to unveiling Miyazaki's ecological vision by understanding the creative processes behind the engaging and symbolic landscapes in his filmography.

More specifically, the research:

- Highlights the ecological vision of Miyazaki, implementing, at the same time, a knowledge on his visual language;
- Systematizes the knowledge of the operative workflows, from analogic to digital, adopted by Miyazaki in the selected films;
- Critically analyzes the visual language employed in the selected Miyazaki's films, illustrating the potential of 2D representations, capable of stimulating the imagination and conveying mental images embedded with symbolic meanings, in contrast to the current trend of developing realistic, tridimensional and immersive visualizations;

- Identifies the character-defining elements of Miyazaki's landscape depictions in order to stimulate environmental awareness, discussing how they could be replicated in contemporary video animation to stimulate critical thinking and reflections on nature and landscape preservation.

The relevance of the present research consists of grasping the approach adopted by Miyazaki in the creation of the representations of natural elements and understanding how he managed to combine them with the plots of his different films, stimulating, indirectly, environmental awareness and a fascination for nature in its different forms.

**Author Contributions:** Although the writing of the methodology and results were shared by the authors, C.B. wrote: 1. Introduction, 3. Materials and methods, 6. Discussion: reflections on landscape representation technique; A.I. wrote: 2. Ecology in Miyazaki's films production; D.M. wrote: Abstract, 4. The creative process: from drawing to video animation, 5. Results: visual poetry for environmental awareness, 7. Conclusions. All authors have read and agreed to the published version of the manuscript.

**Funding:** This research received no external funding.

**Institutional Review Board Statement:** The study did not require ethical approval.

**Informed Consent Statement:** Not applicable.

**Data Availability Statement:** No new data were created.

**Conflicts of Interest:** The authors declare no conflict of interest.

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
