# Peer review of "Representation of Landscape and Ecological Vision in Miyazaki’s Filmography"

_sustainability, doi:10.3390/su152015132_

Round 1
Reviewer 1 Report
The paper, as mentioned by authors, analyzes the central role of landscape in Hayao Miyazaki's films and its representation embedded with symbolic meanings that strongly promoted environmental and sustainability awareness in his production.
This is a well-delineated and properly structured proposal.
The analysis and results are of particular interest.
It is clearly discussed in the paper that the pillars of the visual landscape narrative of Hayao Miyazaki's films achieve a balance of mutual respect between the individual and the genius loci of places in the most metaphysical and spiritual sense. Moreover, for Miyazaki, Man is always at the center of the narrative, and nature and man-made space can be either in harmony (as representing good) or in opposition (representing evil). In the film Castle in the Sky, for example, (as mentioned) nature and urban space (thus, man-made technology) are in constant dichotomy but the consequences associated with the abuse of technology are also shown. Pestilence, destruction, and war are reflected in the desolation of villages, the defensive structures of the castle in the sky, and then in the post-conflict reconstruction of what remains of Howl's castle. A denunciation of human activities that use technology against the landscape is also moved here.
Author Response
see attach

Reviewer 2 Report
Review for: « Representation of landscape and ecological vision in Miyazaki's filmography »
This work presents a study of landscapes in the animated movies from Hayao Miyazaki’s filmography. By studying landscapes, the authors aim at giving proofs of Hayao Miyazaki’s ecological vision. The subject is interesting with a duality between cinematography and environmental awareness. It is however difficult to assess if this article can be considered as an opinion or as a research article, as it lacks some key components. I recommend “Reconsider after major revision”.
Detailed comments:
- While the introduction is interesting, it lacks a review of research articles. Most references in this paper are either non-peer-reviewed books or references to the studied animated movies. Review with references to research articles must be added.
- Figure 12, the house with the red roof is a bad example of direct observation for the movie. In fact, this house was built (2005) after the release of the movie (1988) and is a touristic place reproducing the house in “My Neighbor Totoro”.
- The “Results” part is not really a presentation of results, but a collection of vague “concepts” without clear proofs, such as images with timecodes, analysis on such images or plots. This part needs more work. Also, a part of what is written in the “Results” part should be in the “Discussions” section.
- How can the authors know that Hayao Miyazaki’s animation of elements of nature is a true ecological message instead of a commercial necessity or a story/universe coherence requirement? A discussion about this could be interesting.
- In Hayao Miyazaki’s works, as explained in this article, a duality exists between a traditional rural Japan and a more alienating industrial city side. A part of the traditional Japan is linked to spirituality with the presence of Yōkai and related folklore. It is however a different conception of spirituality compared to the western world’s spirituality. It should be interesting to add some sentences about how the inclusion of Yōkai (and why these specific Yōkai are used in the movie) can be perceived by Japanese people as elements of ecological awareness.
- Also, the context of Hayao Miyazaki’s life needs to be further explored. Hayao Miyazaki was born in 1941 and its vision of life is heavily influenced by the WWII and the post-WWII Japan. This period is very different from ours, and analyses of his animated movies should take this fact into account. Even the ecological vision is different between post-WWII area and nowadays. Note that J.R.R. Tolkien present similar patterns, with the creation of fantasy world and folklore, and an ecological message, following the WWI. More generally, it could be interesting to compare Hayao Miyazaki’s works with other works from similar artists’ profiles.
Author Response
see attach

Reviewer 3 Report
Representation of landscape and ecological vision in Miyazaki's filmography
Please include the results acquired in the abstract
Please list the major significant contributions done in this work by the authors.
Include some related works done by other researchers in this domain and figure out the research gap and show how the gap was filled by your work.
The results should be justified with performance evaluation metrics.
How can you say your results are good?
Please include a discussion section
Try to include a comparative analysis section. And show how your results are better than others.
Include latest references from 2022, 2023
None
Author Response
see attach

Reviewer 4 Report
This paper explores Miyazaki's visual landscape representations and his story-telling on the relationships between humans and nature. It seems like a great article-- but for scholarly article, it lacks substance and novelty, clear implication, and has an unclear copyright issues.
The literature reviews already describes his visual story-telling between human and the ecosystem and does not lead to research question. Half of the citation was only the screen capture of the film. Why do we need to do this research? The method is unclear. How are they analyzed? How are the films selected? Are there existing studies that used this methodology to assess the filmography? Why are synopsis explained? What do the results mean for whom? For the discussion, how do understanding the dynamics and themes of landscapes in these films help anyone in practical or scholarly matter? What can we build or solve from reading this research? These are all unclear.
Most important of all, the images captured are from copyrighted materials by someone else. Do the authors own such rights to use these images? I'm not sure about the laws regarding this-- but the authors should describe or indicate how they have acquired the rights to use these images.
Unless all these significant issues are addressed, the paper should not be categorized as a scholarly article in a scholarly journal.
Author Response
see attach

Reviewer 5 Report
An original, lovingly crafted, and well-structured article. I would especially highlight the strategy of the analysis. Miyazaki's filmography is ideal for highlighting the essential components of what we call landscape, environmental and cultural factors, and in particular their interactions in which perception is also involved. Recommended for publication.
Author Response
see attach

Round 2
Reviewer 2 Report
My comments were addressed. However, a few details should be corrected.
Line 36. It seems that the word "renowned" was duplicated before and after the word "Japanese"
Line 214. Eleven films are mentioned, but the list contains only ten titles.
Author Response
Thank you for pointing this out. We corrected Line 36 and Line 214.
Reviewer 3 Report
The authors have revised the paper but still few comments are not incorporated. Please see the previous comments carefully and revise the paper accordingly.
No Comments
Author Response
Thank you for bringing this to our attention. We have addressed all your comments. We are of course available for any further editing.
Comments 1: Please include the results acquired in the abstract
|
Response 1: Thank you for pointing this out. We agree with this comment. Therefore, we have included the result and the methodology adopted in the abstract. Also, we make more clear the implication of our study. Concerning the copyright issues we already clarify this by email with the editor of the journal.
|
|
Comments 2: Please list the major significant contributions done in this work by the authors. |
|
Response 2: Thank you for pointing this out. We clearly stated the contribution of each author for this publication.
Comments 3: Please include a discussion section Response 3: Thank you for pointing this out. We agree with this comment. Therefore, we have included a discussion session.
Comments 4: Include latest references from 2022, 2023 Response 4: Thank you for pointing this out. We have, accordingly, implemented the bibliography, updating it also with reference from 2023. Also, we provided a more in depth explanation of the results.
Comments 5: Include some related works done by other researchers in this domain and figure out the research gap and show how the gap was filled by your work. Response 5: Thank you for your suggestion. We searched for similar researches that we mentioned in the references and in the text. In the text we clarified the role of our work in bridging the gap with existing studies and researches. Comments 6: The results should be justified with performance evaluation metrics. Response 6: Given the type of analysis we carried out, we were not able to apply evaluation metrics to the word developed. Comments 7: Try to include a comparative analysis section. And show how your results are better than others. Response 7: We did not find any studies that could be used for a meaningful comparative analysis. Therefore, we decided not to report a comparative analysis section.
|
Round 3
Reviewer 3 Report
The authors have addressed all the comments satisfactorily. Hence the paper can be accepted.
Author Response
The Reviewrs 3 clearly stated that: "The authors have addressed all the comments satisfactorily. Hence the paper can be accepted."